# Anthropogenic and Lightning Fire Incidence and Burned Area in Europe

**Jasper Dijkstra** [1,*] , **Tracy Durrant** [2] , **Jesús San-Miguel-Ayanz** [3] **and Sander Veraverbeke** [1]

1   Faculty of Science, Vrije Universiteit Amsterdam, de Boelelaan 1085, 1081 HV Amsterdam, The Netherlands; s.s.n.veraverbeke@vu.nl

2   Engineering Ingegneria Informatica S.p.A., Piazzale dell'Agricoltura, 24, 00144 Roma, Italy; tracy.durrant@ext.ec.europa.eu

3   Joint Research Centre, European Commission, Via Enrico Fermi, 2749, 21027 Ispra, Italy; jesus.san-miguel@ec.europa.eu

*   Correspondence: jasp.dijkstra@student.vu.nl

**Abstract:** Fires can have an anthropogenic or natural origin. The most frequent natural fire cause is lightning. Since anthropogenic and lightning fires have different climatic and socio-economic drivers, it is important to distinguish between these different fire causes. We developed random forest models that predict the fraction of anthropogenic and lightning fire incidences, and their burned area, at the level of the Nomenclature des Unités Territoriales Statistiques level 3 (NUTS3) for Europe. The models were calibrated using the centered log-ratio of fire incidence and burned area reference data from the European Forest Fire Information System. After a correlation analysis, the population density, fractional human land impact, elevation and burned area coefficient of variation—a measure of interannual variability in burned area—were selected as predictor variables in the models. After parameter tuning and running the models with several train-validate compositions, we found that the vast majority of fires and burned area in Europe has an anthropogenic cause, while lightning plays a significant role in the remote northern regions of Scandinavia. Combining our results with burned area data from the Moderate Resolution Imaging Spectroradiometer, we estimated that $96.5 \pm 0.9\%$ of the burned area in Europe has an anthropogenic cause. Our spatially explicit fire cause attribution model demonstrates the spatial variability between anthropogenic and lightning fires and their burned area over Europe and could be used to improve predictive fire models by accounting for fire cause.

**Keywords:** fire cause; burned area; ignition; random forest model; Europe

## 1. Introduction

Wildfires are common in many ecosystems [1]. The major natural cause of wildfires is lightning [2], but humans have increasingly affected fire regimes since their sedentarization [3]. Due to climate change, fire weather extremes such as droughts and heatwaves will become more frequent and put an upward pressure on fire risk [4–8]. As such, some regions on Earth have experienced an increase in fire extent, magnitude and frequency over past decades [9,10].

Lightning and anthropogenic fires are fundamentally different from each other. In the United States, for example, lightning fires tend to be larger and more intense, while anthropogenic fires are more frequent and occur during a larger period of the year than lightning fires [9]. Moreover, the human influence on the spatial and temporal patterns of fires is increasing, such as by promoting drier fuel conditions as a result of land use changes [11].

The attribution of fires to either lightning or human causes has a high spatial variability. This spatial variability is largely explained by the seasonal coincidences between lightning occurrence and low fuel moisture [12] and the human accessibility of landscapes [11]. In

Southern Europe, more than 95% of the fires with a known cause have an anthropogenic origin [13,14]. In central Europe, more than 99% of fires are caused by humans [13]. In mountainous regions, the fraction of lightning fires is higher. In Austria, for example, Müller et al. [15] documented that 15% of the fires were started by lightning, while their relative occurrence increased with altitude. Remote northern landscapes also experience relatively higher fractions of lightning fires. Larjavaara et al. [16], for example, calculated that 13% of fires in Finland were started by lightning.

A better understanding of the spatial variation in ignition causes is imperative to improve or create policies that aim to adapt to or mitigate the detrimental effects of fires. Discriminating between anthropogenic and lightning ignitions positively affected the predictive power of a fire model for Switzerland [17]. Several studies have attempted to identify key wildfire drivers and to what extent they affect the fire regime over parts of Europe [13,17–22]. None of these studies have done this in a spatially explicit manner for Europe while also differentiating between anthropogenic and lightning fires.

Here, we used a fire cause reference dataset from the European Forest Fire Information System (EFFIS) [23] that registered the causes of fires and burned area over many European regions to better understand the drivers of spatial variation in the relative occurrences of anthropogenic and lightning fires. We used this dataset to calibrate and validate a European fire cause attribution model using machine learning. In addition, the model was used to estimate the attribution between anthropogenic and lightning fires for regions where no reference data were available. By doing so, we created the first pan-European fire cause attribution model.

## 2. Data and Methods

### 2.1. Reference Data of Anthropogenic and Lightning Fires and Burned Area

The European Forest Fire Information System (EFFIS) collects data on wildfires from member state countries in the European Union, as well as from other European countries. We retrieved data on the number of wildfires (fire incidence), their causes and burned area from the European Fire Database (EFD) from the EFFIS. In this study, we used data reported between 2001 and 2019, but reported years vary between countries (Table A1). The data are stored at Nomenclature des Unités Territoriales Statistiques (NUTS) level 3 to overcome the differences between different administrative levels between countries. The NUTS regions are statistical subdivisions of countries on the basis of demographic data. NUTS3 is the finest subdivision; however, NUTS3 regions vary in size from approximately 20 km$^2$ in densely populated areas to more than 100,000 km$^2$ in remote areas.

The data in the EFD include records of all known uncontrolled vegetation fires [24]. Different European countries often have their own systems to classify fire causes and, hence, the EFFIS has developed a conversion scheme to harmonize the dataset [25]. Despite these efforts, minor reporting differences between countries may still be present in the EFD.

Since this research attempts to quantify the spatial variation in fire cause, the analysis only included fires with known (anthropogenic or lightning) fire causes, thereby excluding 45% of the fires and 32% of the burned area with unknown cause. Since we were interested in the relative distribution between anthropogenic and lightning fires, we assumed that the fires with unknown causes followed the same distribution as the reported fires. This assumption might be an oversimplification, as authorities might favor investigating probable anthropogenic fire causes at the expense of lightning caused fires due to resource limitations. No fires were reported from The United Kingdom, Belgium, Austria, Serbia, Norway, Albania, Denmark, Ireland, North Macedonia, Iceland, Malta, Liechtenstein, Luxembourg nor Montenegro, which together account for 347 NUTS3 regions. In addition, another 317 out of 1521 NUTS3 regions had not recorded fires, 33 regions only recorded fires with an unknown cause, and 2 registered the fires but not the burned area. All these regions were excluded from the analysis and considered as regions with "no data", so, ultimately, the data from 822 NUTS3 regions were used in the analysis (Figures 1 and 2).

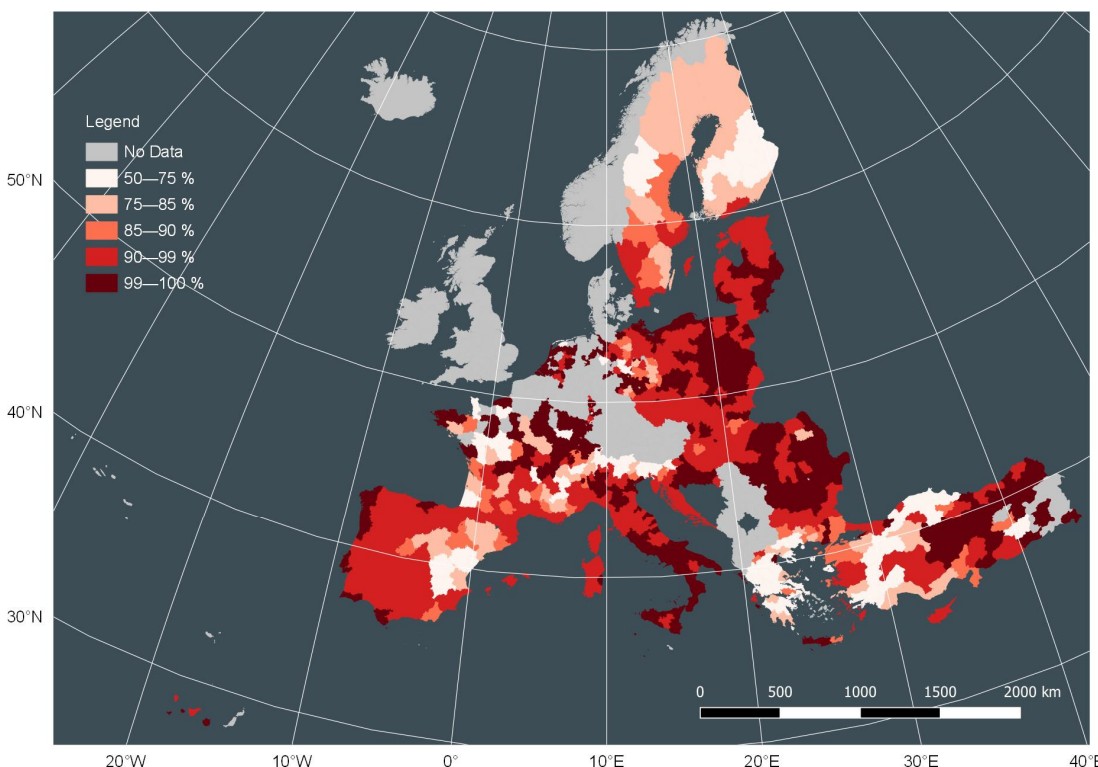

**Figure 1.** Percentage of anthropogenic fire incidence (number of fires) per Nomenclature des Unités Territoriales Statistiques 3 (NUTS3) region based on the available data from the European Forest Fire Information System.

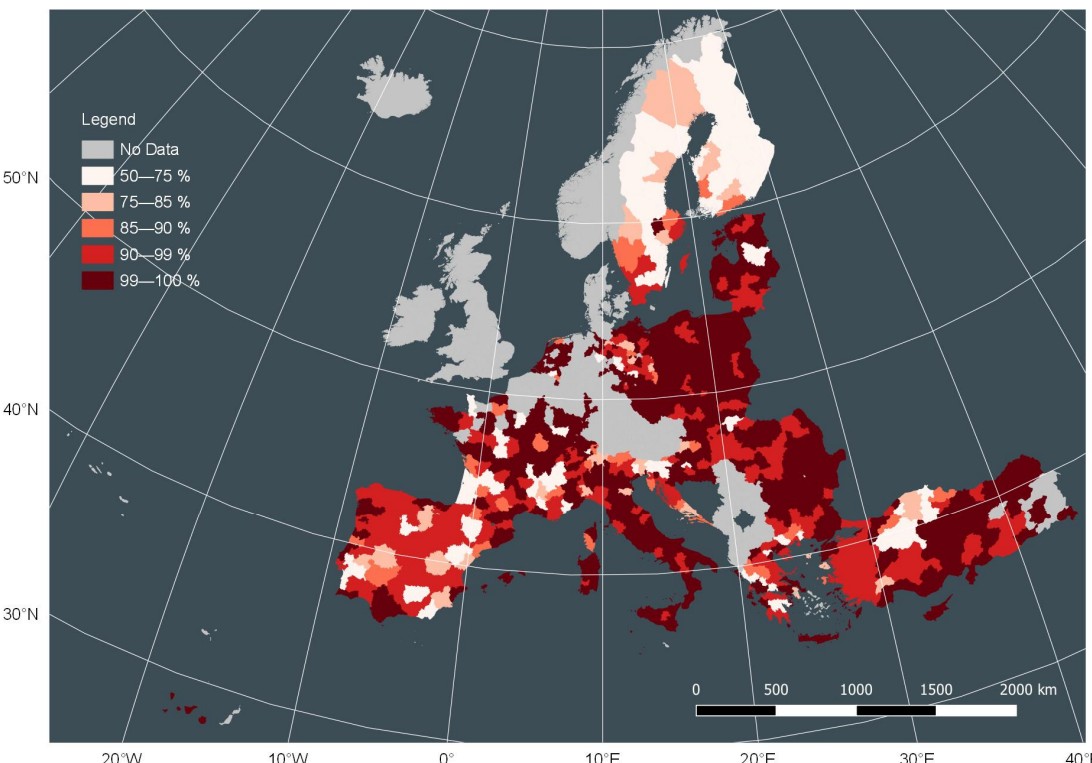

**Figure 2.** Percentage of burned area from anthropogenic cause per Nomenclature des Unités Territoriales Statistiques 3 (NUTS3) region, based on the available data from the European Forest Fire Information System.

## 2.2. Anthropogenic Fire Drivers

Since humans cause the majority of forest fires in Europe, population density was used as a predictor variable. This variable was obtained from Eurostat, the statistical office of the European Union, which offers annual demographic data on several NUTS levels between 1990 until 2019. For our analysis, we calculated the multi-year average within the temporal extent for the NUTS3 regions covered in the EFD.

Since population density is not an inclusive measure of anthropogenic influences on landscapes, we also included the percentage of low impact land [26] as a predictor variable. The low impact land layer dataset was generated at 1 km$^2$ resolution by Jacobson et al. [26] by combining global and spatially explicit datasets on human impacts including ecoregions, land use, nighttime lights, human population, livestock density, forest cover change and areas protected for biodiversity conservation. Jacobson et al. [26] further discriminated between low and very low impact. As such, both layers existed of grid cells of either human impacted (0) or (very) low impacted land (1). We used the low impact layer in our study, and, since we are interested in the human land impact, we calculated the percentage of human impacted land per NUTS3 region as the complement of the low impact land, since low and human impacted land sum to unity. For the 1521 NUTS3 regions in the EFD, the percentage of human impacted land varied between 1% and 100% (Table 1).

**Table 1.** Overview of all environmental variables used in the fire cause attribution model. Summary statistics (Min: minimum, Max: maximum, $x$: mean, $s$: standard deviation) were calculated over the 1521 NUTS3 regions in the European Fire Database from the European Forest Fire Information System.

| Independent Variable | Source | Min | Max | $x$ | $s$ |
|---|---|---|---|---|---|
| Population density [people per km$^2$] | Eurostat | 1.98 | 8927 | 235.9 | 538.7 |
| Human land impact [%] | Jacobsen et al., 2019 [26] | 1.0 | 100.0 | 80 | 18 |
| Lightning flashes per km$^2$ yr$^{-1}$ | Cecil et al., 2014 [27] | 0.13 | 21.32 | 4.36 | 2.78 |
| Burned area coefficient of variation [-] | Giglio et al., 2018 [28] | 0.53 | 12.67 | 4.43 | 4.18 |
| Altitude [m] | Danielson & Gesch, 2011 [29] | −1.22 | 2266.0 | 354.9 | 368.71 |
| Terrain Ruggedness Index [-] | Danielson & Gesch, 2011 [29] | 1.37 | 965.04 | 155.77 | 165.66 |
| Tree cover density [%] | Copernicus Land Monitoring Service [30] | 0 | 64 | 24 | 14 |

## 2.3. Climatic Fire Drivers

We retrieved the mean annual lightning flash rate at 0.5° resolution from the remotely sensed lightning product that combines observations from the Lightning Imaging Sensor (LIS) and the Optical Transient Detector (OTD) [27]. The LIS detected lightning between 1998 and 2014 between 38° N and 38° S, whereas the OTD achieved global coverage between 1995 and 2000. The flash rate estimates over the parts of Europe south of 38° N stem from combining LIS and OTD data, while the flash rate estimates at higher latitudes originate from OTD only. The lightning flash rate has been shown to correlate strongly with cloud-to-ground strikes, which are relevant as potential ignition sources [31]. The mean annual flashes per km$^2$ for each NUTS3 region were calculated as a distance-weighted mean for the NUTS region, centroid from the lightning flash rates of surrounding 0.5° grid cell centers.

Chuvieco et al. [20] established that the interannual variability of wildfires can serve as a proxy for some of the most significant climatic factors that drive forest fires. They described the interannual variability with the burned area coefficient of variation (BA_CV). This metric is defined as the ratio between the interannual standard deviation of the burned area and the annual mean of the burned area over a given number of years.

The BA_CV was computed using data from the MCD64A1 product of the Moderate Resolution Imaging Spectroradiometer (MODIS) [28] between 2001 and 2019, aggregated over the NUTS3 regions. The MCD64A1 dataset has a pixel size of 25 ha, and it is well known that the MCD64A1 product underestimates small fire occurrences [32]. Since a

burned area product that adequately resolves small fires does not currently exist, we used the MCD64A1 product because it captures the spatiotemporal dynamics of fires larger than 25 ha. As high BA_CV values correspond to a lower fire incidence, and vice versa, the BA_CV component can be interpreted as a surrogate for the fire return interval. For the NUTS3 regions where MODIS did not register any burned area, the BA_CV was calculated by using an empirically determined replacement value of the annual mean burned area of $10^{-10}$ km$^2$, as this value approximates zero yet still allows for the calculation of the BA_CV.

### 2.4. Landscape Fire Drivers

Using several input data sources, the United States Geological Survey (USGS) and the National Geospatial-Intelligence Agency (NGA) developed a digital elevation model with global coverage with a spatial resolution of approximately 250 m: the Global Multi-resolution Terrain Elevation Data (GMTED2010) [29]. Because the literature suggests that elevation influences European wildfire patterns [33,34], the GMTED2010 dataset was used to calculate the mean elevation for each NUTS3 region.

Similar to elevation, the heterogeneity of the terrain is another topographic parameter that has been linked to wildfire incidence [35,36]. Riley et al. [37] described terrain ruggedness as the total change between a reference grid cell and its surroundings. Thus, terrain ruggedness is a measure of spatial variation in altitude. Therefore, we used the standard deviation of the elevation of all pixels per NUTS3 region as a proxy of terrain ruggedness.

We also included tree cover as a predictor variable in the analysis. The variable was derived by computing the zonal mean per NUTS3 region from the tree cover map developed by the Copernicus Land Monitoring Service [30], which contains the fraction cover of tree crowns in Europe at a 10 m spatial resolution.

### 2.5. Statistical Analyses

#### 2.5.1. Transformation for Compositional Data

We did not use the absolute number of fires and burned area from the EFD in the analyses, as that would have resulted in inconsistencies between regions because of the differences in temporal coverage. Therefore, we calculated the percentages of anthropogenic and lightning fires and their burned area (Figures 1 and 2). In this calculation, the percentages of anthropogenic and lightning fires amount to 100 percent.

Although the use of percentages or fractions facilitates a comparison between NUTS3 regions, the data cannot directly be used as a response variable in a statistical model. This is because the fractions of lightning and anthropogenic fires sum to one and, as such, are compositional data. This implies that predicting the fractions of anthropogenic and lightning fires separately in a model would not necessarily result in two fractions that amount to a whole [38–40]. To overcome this, the independent variable ($y$) in the statistical analysis was defined using a centered log-ratio (Equation (1)).

$$y = \ln\left(\frac{f_{anthropogenic}}{f_{lightning}}\right) \tag{1}$$

In addition, to overcome problems with undefined ratios and logarithms due to zero values, the anthropogenic fraction was put at $10^{-5}$ when the anthropogenic fraction was zero, and the lightning fraction to $10^{-5}$ when the lightning fraction was zero.

#### 2.5.2. Correlations

We explored the relationships between the anthropogenic, climatic and landscape drivers and the fractions of anthropogenic and lightning fires, and their burned area, using a correlation analysis. In addition to the Pearson correlation coefficient ($r$), we also calculated the Spearman's rank correlation coefficients ($\rho$) to account for non-Gaussian distributions and non-linear relationships.

2.5.3. Random Forest Model

We employed two random forest models in our fire cause attribution modeling. The first model attributes the fraction of anthropogenic and lightning fires, whereas the second one attributes the burned area of anthropogenic and lightning fires. We included only variables in the models that were significant at $p < 0.05$ from the Spearman correlation analysis.

The models were trained on a randomly sampled set of 575 NUTS3 regions. The remaining 247 NUTS3 regions were used to validate the performance of the model. To find the best parameters for each model, 50 different parameter combinations were randomly drawn from a pre-defined parameter space. Each combination was fitted to the training data using 3-fold cross-validation. This process was repeated on a new parameter space, with finer intervals, around the parameter set that yielded the decision tree with the highest coefficient of determination ($R^2$). After that, the process of parameter tuning was repeated for all possible combinations of statistically significant predictor variables. Thereafter, only the variable combinations that yielded the highest $R^2$ scores were considered in the model.

To test the robustness of the model, the variability imposed by the random selection of the training and validation datasets was assessed. Both models were run 100 times for different training-validation compositions. In each run, an extrapolation was done for the NUTS3 zones where no data was reported. Subsequently, the variability per region was computed by means of the standard deviation from the 100 predicted values per NUTS3 region. The modeled outputs were derived from the random forest model that yielded the highest $R^2$ scores in all runs.

Since the model was calibrated to the NUTS3 scale, it is not viable to use it to predict fractions on a national or continental level. The fraction of anthropogenic burned area per country can be calculated as the mean anthropogenic fraction weighted by mean annual burned area per NUTS3 region from the spaceborne MODIS MCD64A1 dataset [28]. Similarly, the fraction of the anthropogenic burned area for the whole of Europe can also be estimated.

## 3. Results and Discussion

### 3.1. Variable Correlations

The correlation analysis between the anthropogenic, climatic and landscape drivers and the fractions of anthropogenic and lightning fires, and their burned area, indicated that they were mostly non-linear, as the Spearman's rank correlation coefficients (Figure 3) outperformed the Pearson correlations. In addition, most relationships exhibited statistical significance at $p < 0.05$. The relationships between the annual lightning flashes and the fire fractions were not statistically significant and, therefore, excluded from the analysis. A possible reason for the absence of a statistically significant correlation is the fact that the lightning flashes dataset is characterized by a strong northward decreasing gradient, which is not as pronounced in the anthropogenic- and lightning-caused fire incidence and burned fractions.

In addition to the dependent-independent variable correlations, the correlation matrix in Figure 3 also reveals some striking collinear relationships between predictor variables, such as the mean altitude and terrain ruggedness index ($\rho = 0.87$). Besides the fact that the terrain ruggedness was derived from the same dataset as the mean altitude, this correlation is a logical result, as terrains will be more rugged if there are more elevation differences. Similarly, more lightning flashes will occur in more rugged terrains ($\rho = 0.52$) due to orographic convection processes [41]. The correlation coefficients also suggest that enhanced human land impact is associated with a decreased tree cover density ($\rho = -0.52$).

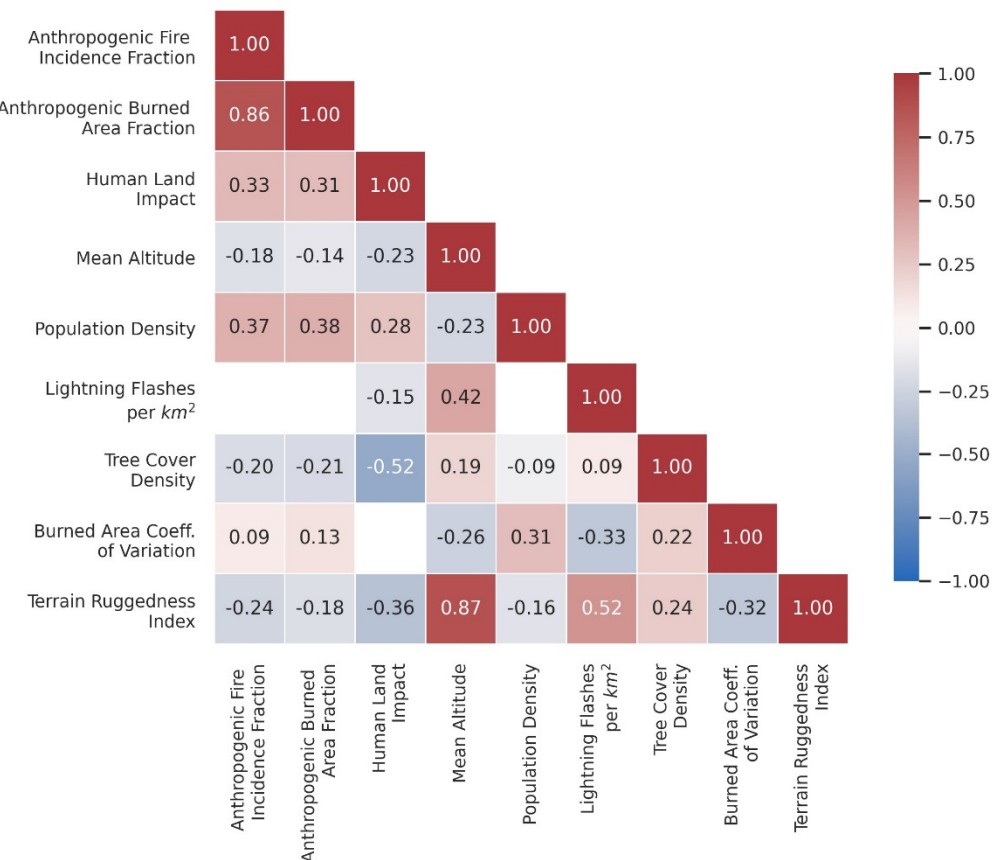

**Figure 3.** Intervariable correlation matrix with Spearman's Rank coefficients ($\rho$) for all variables considered in the analysis. Correlations that were not significant at $p < 0.05$ were left out of the matrix.

Another interesting collinear correlation was found between the burned area coefficient of variation and annual lightning flashes ($\rho = -0.33$). High burned area coefficients are associated with low lightning flash rates. However, as high burned area coefficients relate to a low fire return interval, the trend also implies that there is a positive correlation between the amount of lightning strikes and the fire return interval. The correlation in our data may, therefore, merely be the result of latitudinal climate differences that influence both variables.

A more detailed representation of the statistical relationships between the predictor variables and the anthropogenic fire incidence and burned area fractions revealed clustered relationships (Figure 4). For example, most areas have human land impact fractions larger than 50%, while most anthropogenic fire and burned area fractions are larger than 80%. Similarly, high anthropogenic fire and burned area fractions are associated with lower elevation and terrain ruggedness. Although most low population density values are also clustered to high anthropogenic fire and burned area fractions, there appears to be less scatter in the trend. Low anthropogenic fire and burned area fractions are almost exclusively encountered in regions with low population density, and very high population density almost exclusively in regions where all fires are of anthropogenic origin.

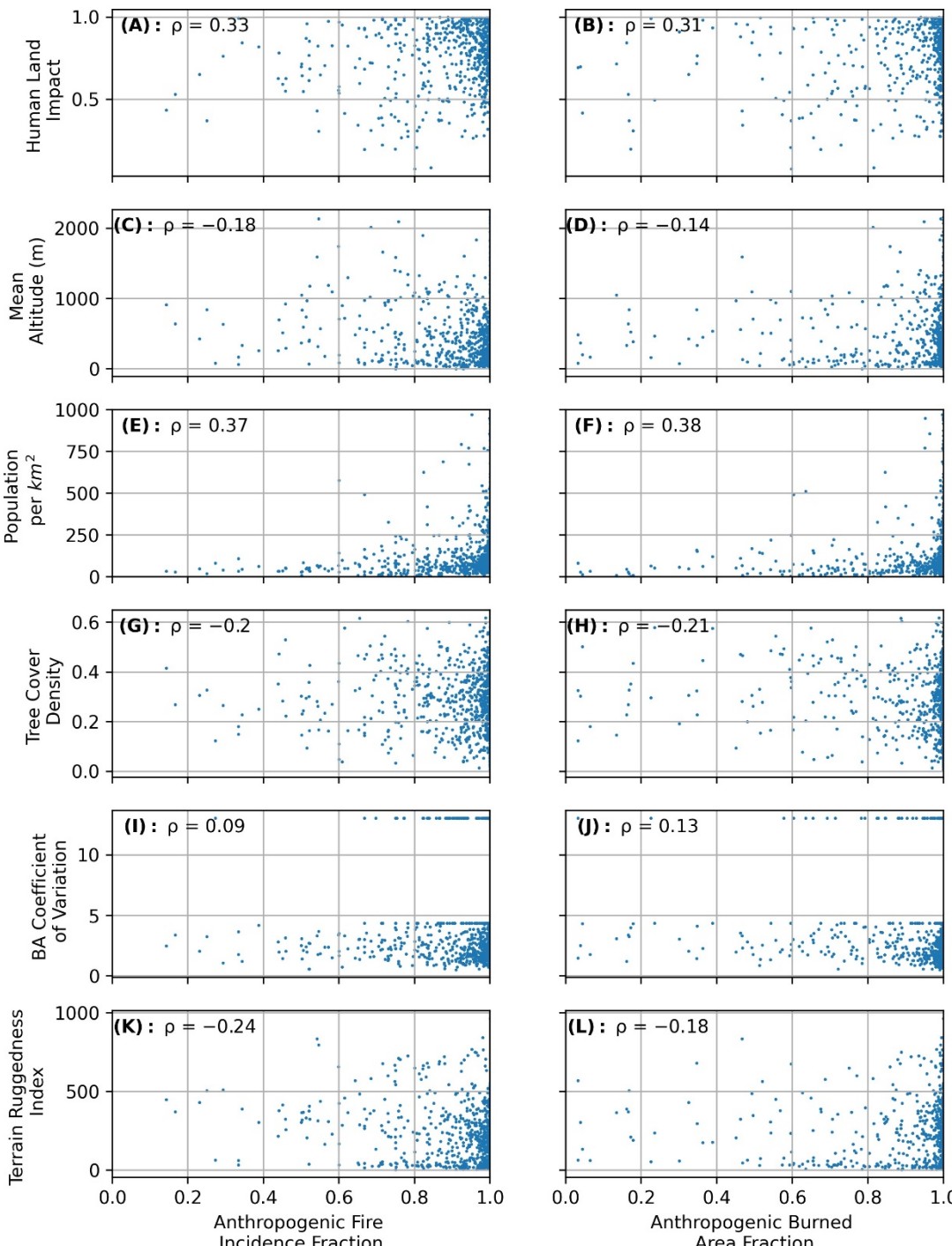

**Figure 4.** Scatter plots with the anthropogenic fire incidence and burned area fractions for each NUTS3 region with known data in the European Fire Database, against the NUTS3 aggregated individual predictor variables: Human Land Impact (**A,B**), Mean Altitude (m) (**C,D**), Population per km² (**E,F**), Tree Cover Density (**G,H**), Burned Area Coefficient of Variation (**I,J**) and Terrain Ruggedness Index (**K,L**).

Unlike the other independent variables, the data of the tree cover density are not skewed. Although the scatter plots in Figure 4G,H are somewhat noisy, they exhibit moderate negative correlations. Thus, high anthropogenic fire incidence and burned area fractions are associated with low tree cover densities. However, as mentioned before, tree

cover density is also strongly negatively correlated to the human land impact ($\rho = -0.52$, Figure 3).

The Spearman correlation coefficients between the fire fractions and burned area coefficient of variation are weak, but statistically significant nonetheless. A probable cause for the low correlation coefficients is the apparent clustering of data points at $BA\_CV$ of around 12 and 4.8, which is caused by imputation values that we implemented for the interpolations in NUTS3 regions where no burned areas were reported within the temporal extent of the EFD. The positive Spearman coefficients imply that the higher the burned area coefficient of variation, and thus the smaller the fire return interval, the higher the fractions of anthropogenic fire incidence and burned area. This is in line with the findings of Catteau et al. [9], who concluded that fire frequency is higher in anthropogenic dominated fire regimes compared to lightning-dominated regimes.

In all relationships, outliers are present. These might be related to data quality issues or because of simplification issues: the variables can have a high spatial variability within a NUTS3 region; however, we used zonal averages in the analysis.

### 3.2. Random Forest Models

After parameter tuning, both random forest models found an optimal $R^2$ when the altitude, burned area coefficient of variation, human land impact and population density were used as explanatory variables. The best $R^2$ that was found after 100 model runs with different train-validate compositions was 0.34 for the fire incidence and 0.35 for the burned area fractions model. The corresponding mean absolute errors (MAE) of the extrapolations of the test data were 0.07 for the fire incidence model and 0.05 for the burned area model. The relatively small MAEs in both models are partly related to the data distribution, which is primarily skewed towards anthropogenic fires and burned area fractions between 0.9 and 1. Therefore, most model predictions also fall within this range and absolute errors remain small.

These performance metrics were based on a single train-validate composition. A more robust measure of model performance is given by comparing the known fire incidence and burned area fractions of each NUTS3 region to the average predicted fractions of the 100 model runs with different train-validate compositions (Figure 5). This shows that, on average, both models have a tendency to overpredict the anthropogenic fire incidence and burned area fractions, as the observations ranged between fractions 0 and 1, while the estimated fractions started around 0.75 for the fire incidence model and around 0.6 for the burned area model.

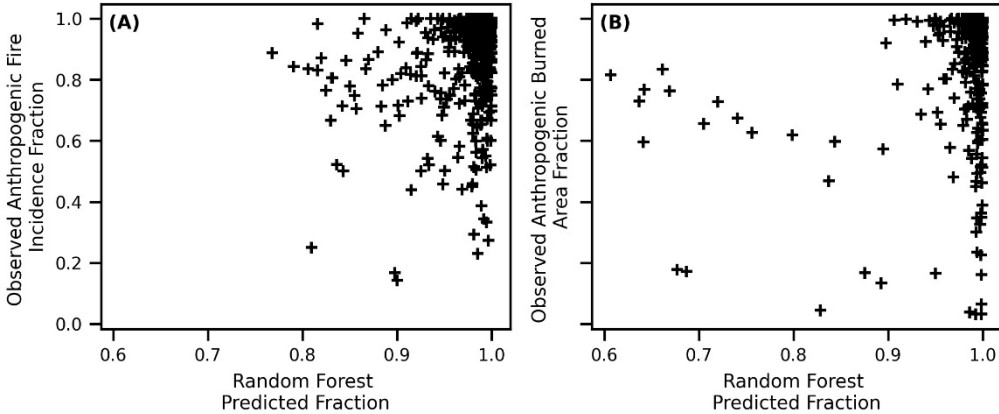

**Figure 5.** The (**A**) observed anthropogenic fire incidence fraction and the (**B**) observed anthropogenic burned area fraction, against the average predicted anthropogenic fraction of 100 random train-validate compositions.

In the fire incidence model that yielded the highest $R^2$, the human land impact variable has the greatest relative contribution in determining the fire incidence fraction (Figure 6A). In the burned area model, the population density is the main contributor to the output fraction (Figure 6B). A likely reason for this difference could be that, regardless of population density, more fires occur in areas with human land impact, while a high population density might be a limiting factor for a fire to grow [42]. In both models, population density and altitude also have a relatively high R-squared, while the burned area coefficient of variation contributes the least to the modelled output.

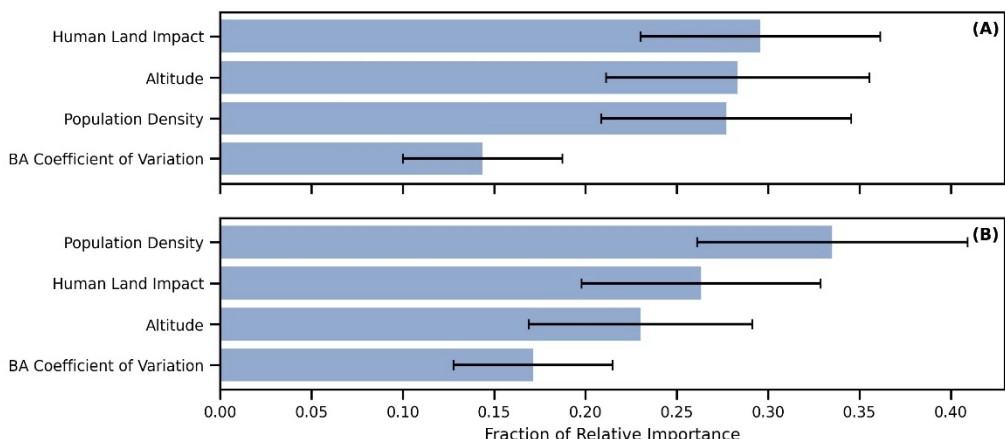

**Figure 6.** The fraction of impurity-based relative variable importance of the predictor variables in the (**A**) anthropogenic and lightning fire incidence random forest model and the (**B**) anthropogenic and lightning burned area random forest model.

A comparable pattern arises when the average loss in $R^2$, due to the exclusion of a predictor variable, is used as a measure of variable importance (Figure 7). The average $R^2$ of 100 model runs with different train-validate compositions was $0.25 \pm 0.04$ for the fire incidence model and $0.24 \pm 0.05$ for the burned area model. When either the altitude or human land impact variables were removed from the fire incidence model, the average $R^2$ of 100 runs decreased to $0.18 \pm 0.045$, while the exclusion of the burned area coefficient of variation caused the $R^2$ to decrease to $0.21 \pm 0.047$. Contradictory to the fire incidence model, the biggest average loss in $R^2$ in the burned area model resulted from excluding the population density variable ($R^2 = 0.18 \pm 0.044$), while the exclusion of the human land impact variable resulted in an $R^2$ of $0.21 \pm 0.041$.

The percentage of anthropogenic fire incidence, modelled by the random forest with the highest $R^2$ score (Figure 8), displays similar geographic patterns as the reference data in the EFD (Figure 1). In remote regions, such as parts of Iceland and Norway, the percentage of anthropogenic fires is lower than in populous areas such as parts of Germany or Denmark. Also in mountainous regions, such as the Austrian Alps, the model predicts lower anthropogenic fire incidence fractions compared to lower elevated areas. Concurrently, the variability in the predictions imposed by the randomness of the selection of the train-validate data is highest in the regions with the lowest anthropogenic fire occurrence (Figure 9). This observation supports the observation of the models' tendency to overestimate the lower anthropogenic fire fractions, inferred from Figure 5.

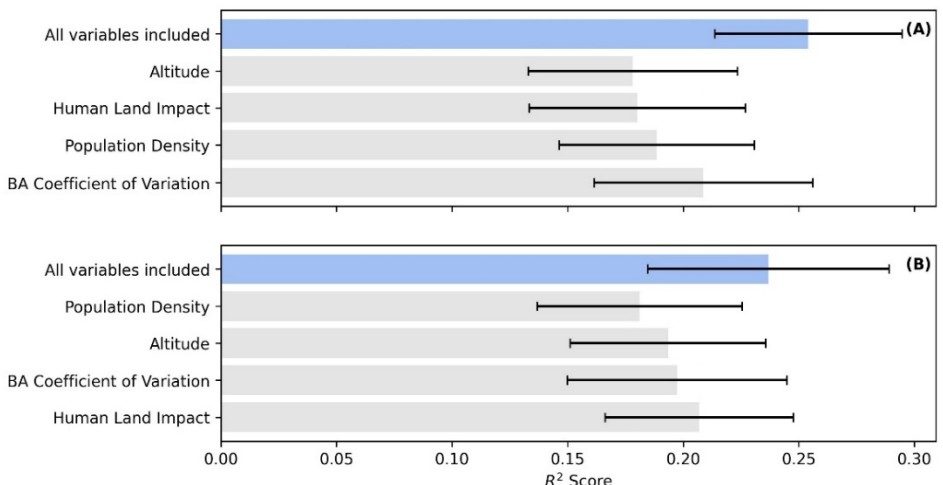

**Figure 7.** The average loss in $R^2$ caused by excluding a predictor variable for (**A**) the anthropogenic and lightning fire incidence and (**B**) the burned area model. This was assessed by determining the mean $R^2$ score of 100 model runs with different train-validate compositions, using the altitude, burned area coefficient of variation, human land impact and population density as explanatory variables (blue bars) and the mean $R^2$ score of 100 different train-validate model runs, excluding one of these variables (grey bars).

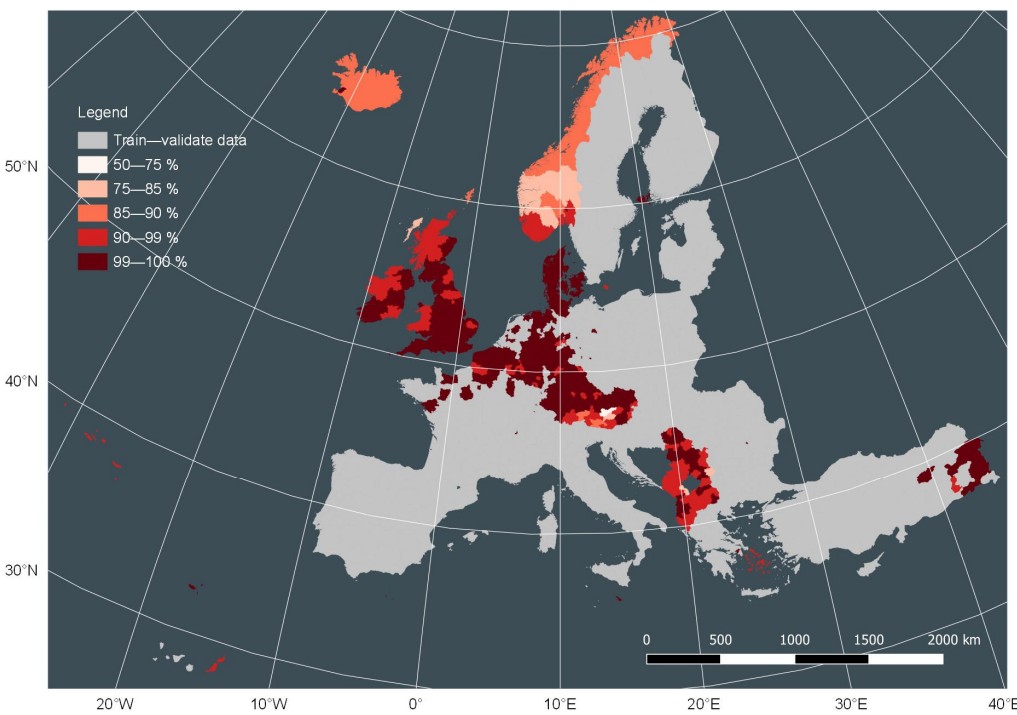

**Figure 8.** Percentage of anthropogenic fire incidence per Nomenclature des Unités Territoriales Statistiques 3 (NUTS3) region extrapolated with a random forest model.

In Western Europe, such as parts of England, Belgium and Germany, the model predicts anthropogenic fire incidences higher than 99%. Although this might appear extreme compared to the surrounding NUTS3 regions in the EFD (Figure 1), the extrapolated regions cover some of the most populous zones of Europe such as, for example, the London metropolitan and Ruhr-Rhine areas.

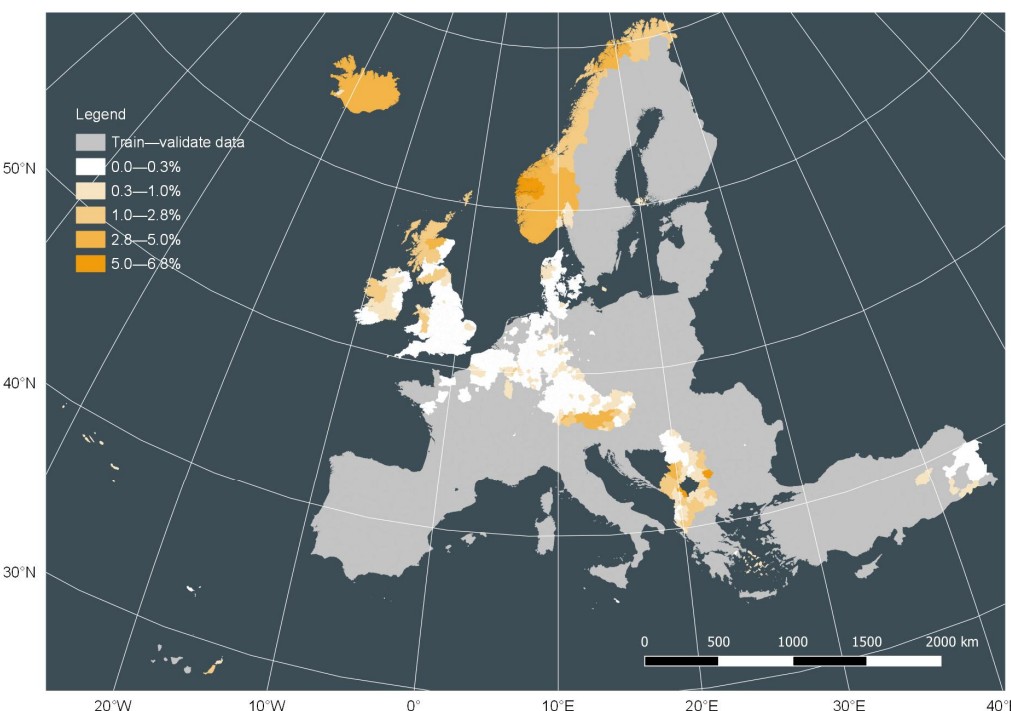

**Figure 9.** Percentage of variation of the extrapolated anthropogenic fire incidence per Nomenclature des Unités Territoriales Statistiques 3 (NUTS3) region. The variability is defined as one standard deviation of all outputs of the model in 100 different train-validate compositions.

The relationship between fire incidence and remoteness is also present in the modelled percentage of anthropogenic burned area (Figure 10). The extrapolated percentage of anthropogenic fire incidence is significantly lower in parts of Iceland, Norway and Scotland compared to, for example, England or Denmark. In addition, the fractions of anthropogenic burned area in these regions appear to be lower than those of fire incidence, which is likely due to the fact that lightning fires are generally larger but occur less frequently than anthropogenic fires [9]. The extrapolation variability imposed by the random selection of the train-validate data is high in remote areas (Figure 11).

In contrast to the fractional fire incidence, a relationship between elevation and the percentage of burned area is not evident from our modelled results. In the burned area fractions retrieved from the EFD, such a relationship is existing, albeit not as obvious as for the fire incidence (Figure 2). Although the relative contribution of the average altitude variable is lower in the burned area model than in the fire incidence one (Figure 6), the prediction variability is somewhat higher in mountainous regions compared to non-mountainous regions (Figure 11). Therefore, it might be possible that the model overestimates the anthropogenic burned area in mountainous areas.

Similar to the anthropogenic fire incidence extrapolations in western and central Europe, the predicted anthropogenic burned area is predominantly larger than 99%. This fits the patterns of the EFD data as well (Figure 2). Both the fire incidence and burned area models are capable of predicting anthropogenic fire incidence and burned area fractions at the NUTS3 scale. The models tend to overpredict the anthropogenic fractions, leading to more variability in the predictions in regions where lightning fires play a significant role.

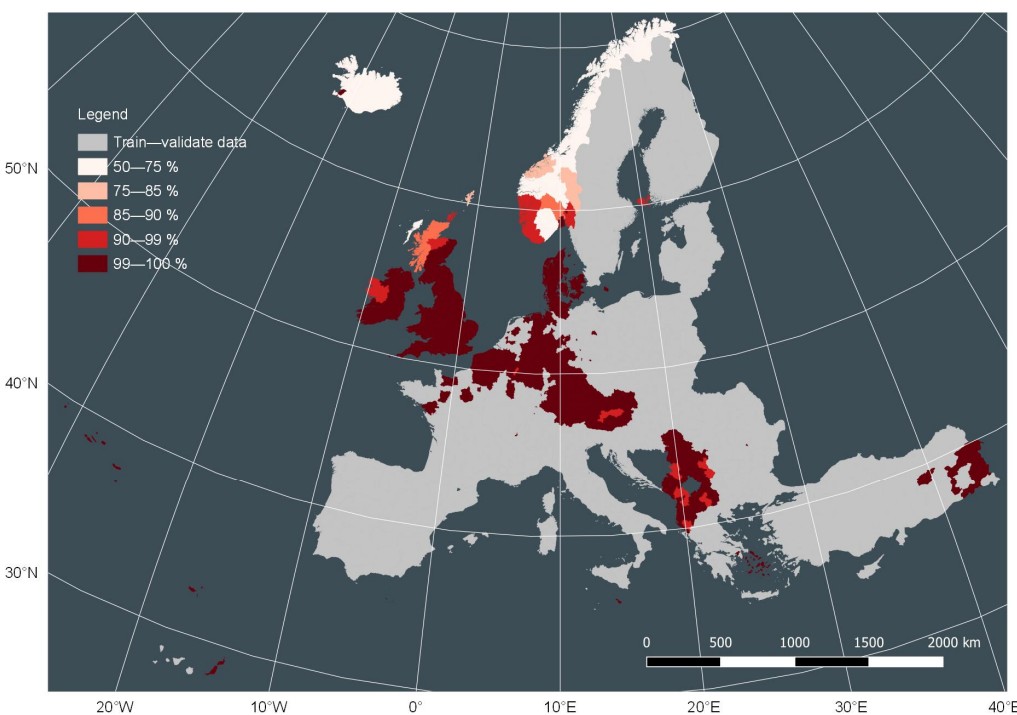

**Figure 10.** Percentage of burned area from anthropogenic cause per Nomenclature des Unités Territoriales Statistiques 3 (NUTS3) region, extrapolated with a random forest model.

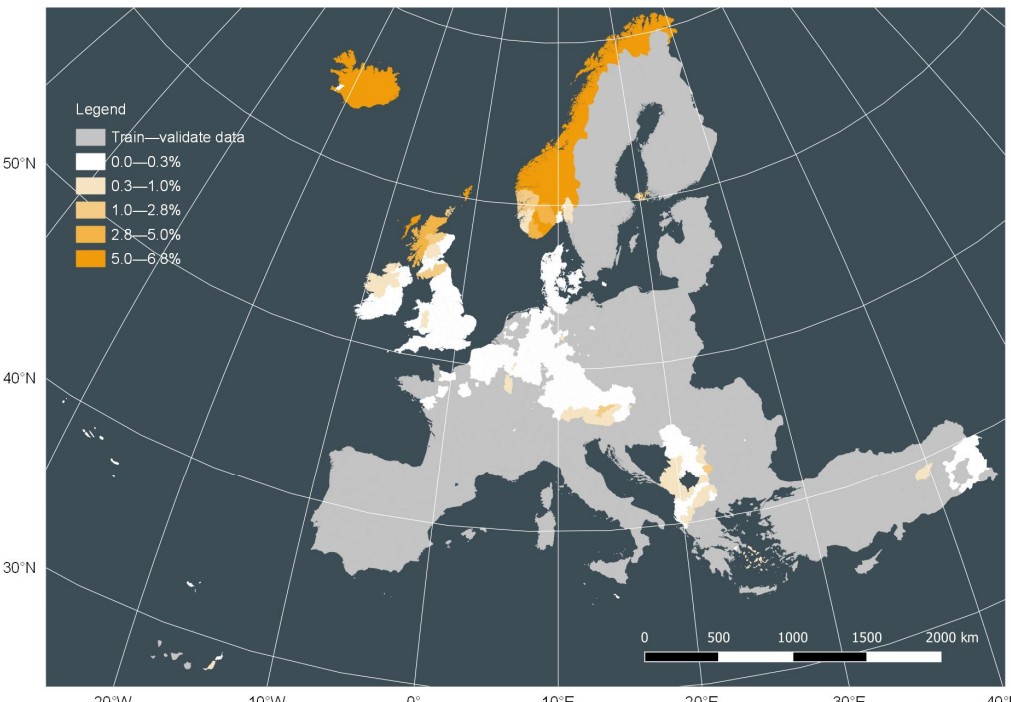

**Figure 11.** Percentage of variation of the extrapolated anthropogenic burned area per Nomenclature des Unités Territoriales Statistiques 3 (NUTS3) region. The variability is defined as one standard deviation of all outputs of the model in 100 different train-validate compositions.

We combined the percentage of anthropogenic burned area per NUTS3 region from the EFD (Figure 2) and the extrapolations of the random forest model (Figure 10) with the MODIS burned area per NUTS3 region to determine the fraction of anthropogenic burned area per country (Figure 12). The percentages of anthropogenic burned area are lowest

in the northern countries Sweden (54%), Finland (67%) and Iceland (70%), while Norway exhibited a higher anthropogenic burned area (87%). Unlike the fractions in Sweden and Finland, which are based on observations from the EFD, the high anthropogenic burned area fraction in Norway might be the result of the overpredicting behavior of the random forest model, since the fraction is solely based on model extrapolations.

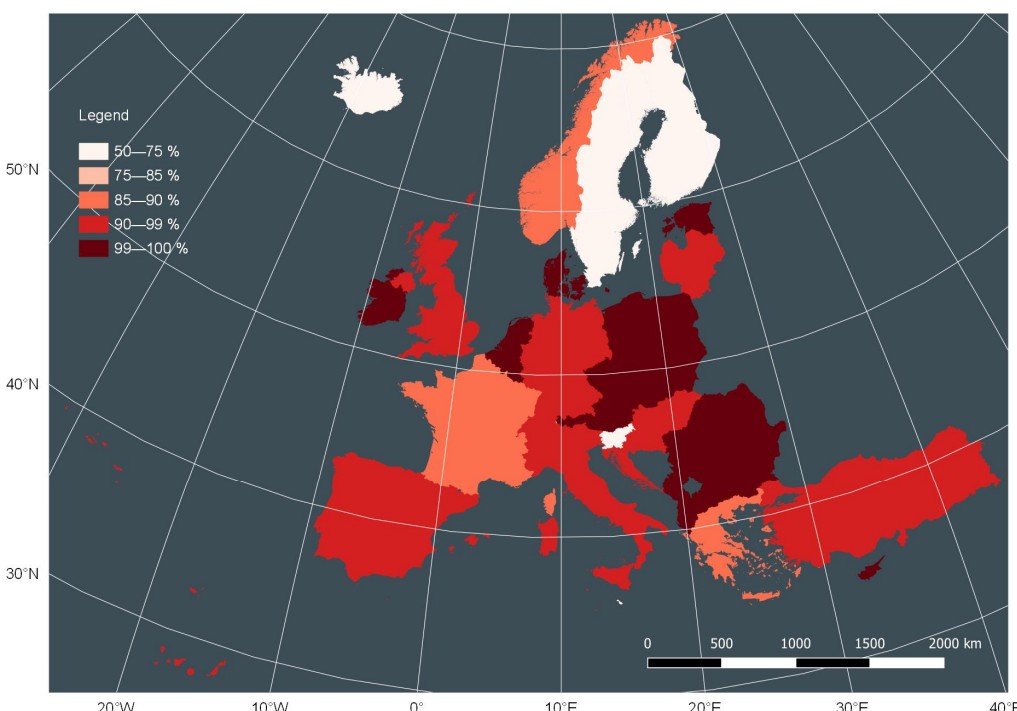

**Figure 12.** Percentage of burned area from anthropogenic cause per country, calculated with the data in the EFD (Figure 2) and the modelled fractions (Figure 10).

Similarly, for Austria, we derived an anthropogenic burned area percentage of 99.7%, while for the neighboring alpine country Switzerland this was 90.9%. Since Austrian observations lacked in the EFD, this percentage was based on modelled values, while the one for Switzerland was predominantly based on data from the EFD. In addition, Müller et al. [15] notes that a significant portion (15%) of fires in Austria is caused by lightning, and Conedera et al. [43] estimated that lightning was responsible for only 4.1% of the burned area in the Alps. Therefore, it is possible that the calculated 99.7% for the anthropogenic burned area fraction in our model for Austria may be an overestimation.

Another potential outlier is Slovenia, where 68% of burned area was attributed to anthropogenic causes. Since the EFD contained data for all NUTS3 regions in Slovenia, this may be related to inconsistencies in data harmonization between different countries in the EFD.

In densely populated countries, such as the Netherlands and Belgium, fires are rare, but, because of easy access, nearly all of the fires and burned area are of anthropogenic origin. Likewise, in eastern European countries, such as Romania, Bulgaria, Poland and the Czech Republic, almost all of the burned area has an anthropogenic cause. This is likely related to land use changes, which have facilitated the formation of large fire prone areas [44].

In France, we estimated that 86.7% of the burned area is due to anthropogenic causes, which is a few percent lower than in surrounding countries. Ganteaume and Guerra [21] showed that wildfire drivers in southeastern France vary spatially and seasonally, depending on the fire source. Therefore, it is likely that the relatively low fraction of anthropogenic burned area in France, is a consequence of the high diversity in climate, land use, topogra-

phy and socio-economic conditions in the country, which includes remote mountainous regions with prevalent lightning fire occurrence.

Based on the MODIS burned area data across Europe and the observations and extrapolations for all NUTS3 regions in the EFD, we estimated that $96.5 \pm 0.9\%$ of all the burned area in Europe is from anthropogenic origin.

### 3.3. Model Performance

The $R^2$ scores of the models that we selected for the extrapolations suggest a reasonable fit between the train and validate datasets. The model performance was partly influenced by external factors such as the selection of predictor variables. Our final models only included four variables, as the selected combinations yielded the best scores, and we wanted to limit the models' loss in predictive power as a result of collinearity among the variables. Despite including the most prominent anthropogenic, landscape and climatic drivers of causes of fire in our model, it may be possible that other variables may further enhance model performance. One particular variable of interest could be the seasonal correlation between lightning and burned area [45], as this proxy may further constrain the likelihood of lightning fires.

Another external factor impacting the $R^2$ score of the model is the quality of the input data. Despite the harmonization of the data in the European classification scheme, reporting inconsistencies between countries persist. The fraction of anthropogenic fire incidence in most Greek NUTS3 regions, for example, is markedly lower than in surrounding Mediterranean countries (Figure 1). The same goes for the anthropogenic burned area fraction in Slovenia (Figures 2 and 12). There is, however, no indication that fire patterns in these countries are indeed anomalous compared to neighboring countries. In addition, the data in the EFD is built up of empirical observations, suggesting a connection to terrain accessibility as well. Fires in areas that are difficult to access, which will predominantly be caused by lightning, will, therefore, more likely pass unnoticed or their cause will be harder to retrace, than it would in highly accessible areas. Moreover, as noted before, the actual cause of fires that have been classified as unknown might have been uninvestigated rather than being undetermined. This might have impacted the ratio between anthropogenic and lightning induced fires in some NUTS3 regions and could partially explain the overestimating behavior of the model.

In addition, the data in the European Fire Database is recorded at the administrative NUTS3 scale, which means that the subdivision of the regions is based on population density, rather than equal surface area. Therefore, remote NUTS3 regions often cover a large area. Since the data of the predictor variables were aggregated to the NUTS3 scale as well, a large variation in anthropogenic, climate and landscape drivers within a zone will get lost in the aggregation process. This problem becomes more prominent when the area of a NUTS3 region gets larger. Therefore, it will be harder for the model to make precise predictions for large NUTS3 regions.

The composition of the train and validate data sets also affected the model performance (Figures 9 and 11). Each composition is randomly drawn from all data in the EFD, but only 17% of NUTS3 regions in the EFD have reported a lightning induced burned area of more than 10%, and 23% of the regions had a lightning fire incidence of more than 10%. Therefore, it is likely that regions with relatively high lightning fires and burned area fractions will be underrepresented in a random train-validate selection. Correcting for the imbalance between anthropogenic and lightning fires in the statistical approach may further improve the model performance.

### 3.4. Implications and Directions for Future Work

Although our predictive models capture some important spatial heterogeneity of anthropogenic, climatic and landscape drivers, model performance could be further improved by, for example, including more training data in the model, especially from regions dominated by lightning fires. Moreover, the current model provides a static view of the

anthropogenic and lightning induced fire incidence and burned area fractions, but it might be interesting to expand the model with a temporal component. By running the model for several time windows, the change in anthropogenic and lightning fire and burned area fractions over time could become apparent. Similarly, the effects of future changes in population density, human land impact or burned area coefficient of variation on the ratio between anthropogenic and lightning induced fire incidence and burned area could be tested. Yet, such analyses would require changes in data structure and reporting consistency of the European Fire Database.

Although the model that we developed in this project has been trained to data in Europe, it might also be useful for other regions in the world on an administrative level similar to the NUTS3. This means that we developed a model that is able to predict the fractions of anthropogenic and lightning induced fire incidence and burned area at a regional scale. In addition to this, the burned area fractions can be used in combination with regionally aggregated MODIS burned area data to determine the anthropogenic and lightning induced fire fractions at, for example, national or continental scales.

## 4. Conclusions

We created a pan-European attribution between anthropogenic and lightning fires and their burned area. We, therefore, used reference data from the European Fire Database where available, and we used this data to develop two random forest models to predict the fractions of anthropogenic and lightning induced fire incidence and burned area at the NUTS3 level for regions without reference data. Our models achieved reasonable performances with $R^2$ scores of 0.34 and 0.35 from regressions between estimated values and independent validation data. The models tended to overestimate the anthropogenic fractions in regions where lightning fires are more important. Despite this shortcoming, our models successfully captured spatial variability in anthropogenic, climatic and landscape drivers of fires such as the higher importance of lightning fires in more remote regions, such as Scandinavia or the Alps.

Both the model results and the observations registered in the EFD showed that, even in the more remote regions, humans are the predominant source of fires and burned area in Europe. We estimated that $96.5 \pm 0.9\%$ of the burned area in Europe has an anthropogenic origin. At a regional (NUTS3) scale, the main driver behind the fraction of anthropogenic and lightning induced fire incidence was the fraction of human impacted land, while the main driver of the burned area fractions was population density. Our model results can be useful to optimize predictive fire models by accounting for fire cause when projecting future fire regime changes under changing climatic and socio-economic conditions in Europe.

**Author Contributions:** Conceptualization, J.D. and S.V.; data curation, J.S.-M.-A., T.D. and J.D.; formal analysis, J.D.; funding acquisition, S.V.; investigation, J.D.; methodology, J.D. and S.V.; project administration, J.D. and S.V.; resources, J.D. and S.V.; software, J.D.; supervision, S.V.; validation, J.D. and S.V.; visualization, J.D.; writing—original draft preparation, J.D.; writing—review and editing, S.V., J.S.-M.-A. and T.D.; All authors have read and agreed to the published version of the manuscript.

**Funding:** S.V. acknowledges the support from the Dutch Research Council through Vidi grant 016.Vidi.189.070 and from the European Research Council under the European Union's Horizon 2020 research and innovation program (grant agreement No 101000987).

**Institutional Review Board Statement:** Not applicable.

**Informed Consent Statement:** Not applicable.

**Data Availability Statement:** The datasets used as predictor variables in our study are publicly available. An overview of these datasets and their sources are displayed in Table 1. An Esri Shapefile containing both the input and modeled fire incidence and burned area fractions per ignition source per NUTS region has been archived in a publicly available GitHub repository (https://github.com/jasper-dijkstra/MSc_ResearchProject, accessed on 27 February 2022). In addition, this repository contains all codes, raw output data of the models (comma separated files) and figures included in

this article. All analyses have been done with the programming language Python (version 3.7.10) (https://www.python.org/, accessed on 27 February 2022).

**Conflicts of Interest:** The authors declare no conflict of interest.

## Appendix A

**Table A1.** Overview of the time windows in which countries submitted data to the European Fire Database.

| Code | Country | From | To | Notes |
|---|---|---|---|---|
| AL | Albania | - | - | |
| AT | Austria | - | - | |
| BE | Belgium | - | - | |
| BG | Bulgaria | 2005 | 2019 | |
| CH | Switzerland | 2001 | 2018 | |
| CY | Cyprus | 2001 | 2016 | Only Greek part of CY |
| CZ | Czech Republic | 2004 | 2019 | |
| DE | Germany | 2001 | 2018 | Some NUTS regions missing |
| DK | Denmark | - | - | |
| EE | Estonia | 2005 | 2018 | |
| EL | Greece | 2001 | 2011 | Some NUTS regions missing |
| ES | Spain | 2001 | 2015 | Some NUTS regions missing |
| FI | Finland | 2005 | 2019 | |
| FR | France | 2001 | 2018 | |
| HR | Croatia | 2001 | 2019 | |
| HU | Hungary | 2002 | 2019 | |
| IE | Ireland | - | - | |
| IT | Italy | 2001 | 2015 | Autonomous regions (e.g., Sicily, Sardinia) often missing |
| LT | Lithuania | 2004 | 2018 | |
| LI | Liechtenstein | - | - | |
| LV | Latvia | 2004 | 2018 | |
| ME | Montenegro | - | - | |
| MK | North Macedonia | - | - | |
| MT | Malta | - | - | |
| NL | Netherlands | 2017 | 2018 | |
| NO | Norway | - | - | |
| PL | Poland | 2001 | 2018 | |
| PT | Portugal | 2001 | 2018 | |
| RO | Romania | 2004 | 2018 | |
| RS | Serbia | - | - | |
| SE | Sweden | 2001 | 2018 | 1242 fires (451 ha) added to "Unknown code" category, as local and EU codes are mutually inconsistent |
| SI | Slovenia | 2001 | 2019 | |
| SK | Slovakia | 2004 | 2018 | |
| TR | Turkey | 2005 | 2013 | 2009 and 2011 missing |
| UK | United Kingdom | - | - | |

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
