# Peer review of "Anthropogenic and Lightning Fire Incidence and Burned Area in Europe"

_land, doi:10.3390/land11050651_

Round 1
Reviewer 1 Report
I found this an interesting article on an important topic. While lightning is not the major cause of wildfire ignition in Europe, it is still significant especially in more remote areas, as the authors point out. In addition, the approach used by the authors is of interest in other regions of the globe where lightning is a very substantial cause of wildfire ignitions (e.g. the overwhelming majority of fires in the Australian Black Summer were ignited by often extensive dry lightning activity).
My approach to this manuscript is from the perspective of an earth system scientist rather than a machine learning specialist nonetheless I found the description of the background data, methods and results clear and readily understandable. This is important, as many of the article's readers will have a similar background, I suggest.
I recommend acceptance for this article. I could identify only a few points that require clarification, and only minor typographical or stylistic issues, as detailed below.
Suggestions for change include:
L.41: "The attribution between" - sounds awkward, suggest something like "The attribution of fires to either lightning or human causes..."
L.45: "higher"? - 15% of fires in Austria attributed to lightning is still not high.
L.57: "dataset" instead of "data"
L.158-9: probably worth clarifying "European wildfire patterns" - both references are explicitly European (and reasonably so, given the focus of the article)
L.239-242: interesting! worth a reference (for non-meteorologists) confirming the statement about enhanced convective activity in rugged terrain> this (quickly found) is an example: Kottmeier, C., Kalthoff, N., Barthlott, C., Corsmeier, U., Van Baelen, J., Behrendt, A., Behrendt, R., Blyth, A., Coulter, R., Crewell, S. and Di Girolamo, P., 2008. Mechanisms initiating deep convection over complex terrain during COPS. Meteorologische Zeitschrift, 17, pp.931-948.
L.266: "implies" - is it not the case that the relationship between anthropogenic fire and low tree cover densities is explicitly represented in the plot? If so, that should be presented as: "Thus, high anthropogenic fire incidence and burnt area fractions are..."
L. 293 remove comma after "compositions"
L.312-316: worth a comment that population density and altitude also had relatively high R-squared.
L.334-337: I suggest that it is "likely" more than "possible" - there is literature especially in North America indicating that remote fires tend to grow larger than those close to settlements due to the difficulty of deploying fire suppression resources to the former.
L.360: "such as" instead of "like"
L.450-464: interesting and detailed discussion - thank you!
L.492-498: the development of the ignition module for the Australian Fire Danger Rating System has some parallels to this work, and could potentially benefit from it.
Reviewer 2 Report
Review of Dijkstra et al. (2022) - Anthropogenic and lightning fires incidence and burned area in Europe.
This manuscript by Dijkstra et al. proposes a random forest model for predicting the fraction of anthropogenic and lightning fire incidences, and respective burned area, at the regional scale (NUTS3) for Europe. The model is calibrated with the reference values obtained from the Copernicus database (EFFIS). The predictors are the variables among population, physical and orographic factors. The manuscript addresses a topic of extreme importance for Europe, a densely populated region with significant wildfire events in recent years and fits well within the scope of the journal. The original study is relevant to our knowledge and understanding of the origins of wildfires and possible ways to mitigate them. The manuscript is clearly written and well-structured with quality from the scientific point of view.
I would recommend this manuscript to publishing after minor revisions. My revisions are briefly and consists in adding simple sentences/paragraphs to the text.
This review is divided into two parts, the first part with main comments and a second part, point-by-point, where small typos, spelling errors and minor changes to the text will be pointed out.
In the manuscript it is necessary:
General comments
The summary is clear and well structured. The problem is framed, and the results are briefly presented, although nothing is said about the methodology used or the models employed. Please, clarify this point.
The literature review is timely and relevant to the study but is limited to a few examples from central Europe (it is intended for all of Europe), and nothing is said about similar studies that using other models or methodologies reaching similar conclusions. I suggest adding references (I leave two suggestions, but the authors can use others) with examples for southern Europe. It is impossible to write an article on wildfires in Europa and do not mention the Mediterranean region, where wildfires fires are a determining feature for the region.
Aparício, B. A., Pereira, J. M., Santos, F. C., Bruni, C., & Sá, A. C. (2022). Combining wildfire behaviour simulations and network analysis to support wildfire management: A Mediterranean landscape case study. Ecological Indicators, 137, 108726.
Parente, J.; M.G. Pereira; M. Amraoui; F. Tedim. "Negligent and intentional fires in Portugal: Spatial distribution characterization". Science of The Total Environment 624 (2018): 424-437. https://doi.org/10.1016%2Fj.scitotenv.2017.12.013. 10.1016/j.scitotenv.2017.12.013
Minor Comments
Line 33 – size? Perhaps “extent”
Line 36 – United States -> United States,
Line 87 - this->This
Reviewer 3 Report
Minor comments:
- Lines 93-94: What does "822 statistical regions" mean? Are they different from NUTS3 regions?
- Table 1: Are there regions with 0 and 100% human land impact? (Taken from the Min and Max columns).
- Line 152: Why did the authors choose such a value for the annual mean burned area for regions without any burned area?
- Lines 241-242: Regarding collinearities, it may be interesting to talk about -0.52 Spearman's Rank coefficient between Tree cover Density and Human Land Impact, which is later mentioned in lines 267-269.
- Figure 6: The authors should consider organizing the predictor variables from more to less relative importance.
- Lines 332-338: Authors repeat information from lines 312-316. They should consider unifying those two paragraphs.
- Minor typos are present, like line 417: "that that."
Major comments:
- Lines 82-94: It is unclear if the 347 NUTS3 regions that had not reported fires are a subset of the 664 NUTS3 regions that had not recorded fires. Are those two groups represented in Figures 1 and 2 as "No Data"? Also, what is done for the 33 regions that only recorded fires with an unknown cause and the two regions that registered only fires and not burned areas? To have a clear perspective, the authors should consider clarifying all those aspects in the text and the total number of regions.
- Lines 117-119: Is that calculation assuming that land impact is either low or human? Could the authors clarify how the percentage of human-impacted land is calculated from the low impact layer?
- Line 185: The problems are not with logarithms with zero values (when f_anthtropogenic=f_lightning). The problems are with ratios with zero values that cause the logarithm to become undefined.
- Section 3.2: Have the authors considered using techniques to overcome the unbalance problem in the data, which relates to the issue explained in the last paragraph of section 3.3.
- Uncertainty in the statistical analysis: In the text, the word uncertainty is used in a way that can lead to confusion. The computed uncertainty only includes the variability caused by running different random forest instances. However, there are many sources of uncertainty in a machine learning model. Indeed, the very low R^2 presented suggests that the predictions are not very reliable since the best model performance is missing 66% and 65% of the variability present on the targets. Therefore, plotting some regions with an uncertainty of 0-0.3% could lead to confusion. In my opinion, the word uncertainty should be replaced or more accurately presented.
- Section 3.3: In my opinion, the authors are being somehow optimistic, stating that they achieved moderate model performance. The 34% and 35% variability in anthropogenic fire incidence fraction and anthropogenic burned area fraction explained from the predictors in the best models shows some big limitations. More predictors seem to be needed to explain some of the variations among regions. In addition, some of the reasons given in consequent paragraphs should be further analyzed using the data. In my opinion, the temporal aspect explained in section 3.4 is key and should be part of the paper in the first place.
Round 2
Reviewer 3 Report
The authors have fixed or considered all my observations deeply, and the text, in my opinion, is ready to be published.
Author Response
Dear reviewer,
Thanks for your comments, they have been very helpful to improve our manuscript.
Further, at suggestion of the editor, we have expanded section 3.3 by proposing that the inclusion of the seasonal correlation between lightning and burned area might be of interest to improve the performance of the model:
“Despite including the most prominent anthropogenic, landscape and climatic drivers of causes of fire in our model, it may be possible that other variables may further enhance model performance. One particular variable of interest could be the seasonal correlation between lightning and burned area [45] as this proxy may further constrain the likelihood of lightning fires.” (L.458-L.462)
ref [45]: https://agupubs.onlinelibrary.wiley.com/doi/abs/10.1029/2020RG000726
Yours Sincerely,
Jasper Dijkstra and Dr. Sander Veraverbeke